# Research on Population Development in Ethnic Minority Areas in the Context of China's Population Strategy Adjustment

**Jinwei Huo [1,*], Xinhuan Zhang [1], Zhiping Zhang [1,2] and Yaning Chen [1]**

[1] State Key Laboratory of Desert and Oasis Ecology, Xinjiang Institute of Ecology and Geography, Chinese Academy of Sciences, Urumqi 830011, China; zhangxh@ms.xjb.ac.cn (X.Z.); zhangzhiping15@mails.ucas.ac.cn (Z.Z.); chenyn@ms.xjb.ac.cn (Y.C.)

[2] The Graduate School, University of Chinese Academy of Sciences, Beijing 100049, China

[*] Correspondence: huojinwei@ms.xjb.ac.cn; Tel.: +86-0991-7827315

**Abstract:** Against the background of China's relaxation of family planning standards, this thesis analyzed the demographic trends in ethnic minority areas and their impacts on regional development under China's adjustment of its population strategy. By setting up different fertility scenarios, the population forecasting software (PADIS-INT) was applied to forecast the population scale and structure of the Hotan region. This thesis analyzed the impacts of population growth on regional sustainable development from the perspectives of employment, economic development, and resource carrying capacity to provide references for the formulation and implementation of population and economic development policies in minority areas, to alleviate the contradiction between the human and environment. The results showed that the Hotan region would maintain a relatively fast population growth rate for a long period; by 2050, its population would skew younger when compared to China's general statistics. However, due to the lagging economic development and the constraints to resources and the environment, unemployment would become the most severe problem hampering regional development. While developing its local economy, the Hotan region needs to better promote the interregional migration of the labor force.

**Keywords:** population development; labor force; employment; resource environment; ethnic areas

## 1. Introduction

In the context of global change, with the increase in the size of the population and its material and cultural needs, the intensity of human activities and their impact on the society, economy, and ecosystems have increased, and the internal conflicts in the human-environment relationship have become increasingly prominent [1–3]. The correlation between population and resources and environment is a central issue for regional sustainable development [4–6]. Humankind has long recognized that the Earth has limited space and resources. Humans have also realized the importance of living in harmony with nature, making great efforts to learn about their surrounding resources and environment [7]. The survival and development of human beings need to obtain sufficient means of production from the resource and environment system. However, under certain technological conditions, the demand of human beings exceeds the supply of the resource and environment system, which will lead to the deterioration of the environmental system and endanger the survival and development of human beings. Maintaining the population within the carrying capacity becomes the scale to measure the state of regional sustainable development. The Malthusian theory of population [8], Schultz's human capital theory [9], and the demographic dividend theory

of Bloom et al. are all concerned with the effects of population growth externally [10]. The Limits to Growth, published in 1972 by the Club of Rome, first defined the resource and environmental carrying capacity: Rapid industrialization, soaring population growth, food privatization, depletion of non-renewable resources, and degradation of the ecosystem. At some point, global growth will hit its limit due to food shortages and ecological devastation. That limit is the carrying capacity [11]. As its population continues to grow and socioeconomic development accelerates, problems such as resource shortages and environmental pollution have become increasingly important constraints to sustainable development in some parts of China. China has launched a series of scientific and technological projects targeting the north-west and the north-east regions to study the supporting capability and carrying capacity of resources and environment for each region's society and economy [12–18]. Studies have also been conducted on the supporting capabilities and protection levels of the resources and environment for each region's population and economy, pointing out that the increased vulnerability of the environmental system caused by the increase in human activities has become the main obstacle to the sustainable development of the regional ecological environment and social economy, and proposing measures to implement ecological migration, developing carbon sink industry, and protecting ecological water. The goal is to provide a scientific basis for the realization of regional resources and environmental security, as well as the sustainable development of the population's socioeconomic conditions [19–24].

To maintain the balance between population and resource environment, China has a long history of prioritizing population control. Since the implementation of family planning policies, 400 million fewer people have been born in China. In the early-1990s, the total fertility rate declined to a level far less than the replacement and has continued to decline, resulting in a fertility rate of 1.5 [25,26]. This has effectively alleviated the pressure that the population imposes on the environment and resources; however, the long-term low fertility rate has led to an ageing problem. Even after the Chinese government relaxed birth restrictions, the total fertility rate (TFR) continued to show a downward trend. An excessively low fertility rate can have a negative impact on socioeconomic development. As China's socioeconomic development enters a new era, China has shifted its population strategy from merely controlling the number to addressing specific issues (including population scale, structure, and quality) in a coordinated manner. The population strategy has also been adjusted from addressing internal problems of the population system to rational interactions with the external environment [27]. However, situations in ethnic minority areas are different from those in China as a whole. Due to preferential policies for ethnic minorities, family planning is already relatively loose in ethnic areas with relatively rapid population growth. Moreover, since most of China's ethnic minority regions are located in impoverished places (such as the frontier and drought-stricken areas) where economic development lags and productivity levels are low, increasing the labor force has been the main way out of poverty. Inhabitants in ethnic minority areas generally share a strong will to have children which has resulted in a population growth rate much higher than the national average. However, when the total population exceeds the carrying capacity of the region's resources and environment, it starts to cause unemployment and environmental crises. The population problem has become a critical constraint for the sustainable development of China's ethnic areas and the realization of poverty alleviation. However, there are few studies on the impact of the adjustment of family planning on population development in ethnic minority areas. On the southern edge of the Taklamakan Desert, Hotan in Xinjiang features a fragile ecological environment; it is a minority area located on the border and a contiguous area of extreme poverty. It is a typical "frontier, minority and impoverished" region in China. This case study used the example of the ethnic minority area of Hotan. The hypothesis was the abolishment of the family planning policy. By setting up different fertility scenarios, the PADIS-INT was applied to forecast the population scale and structure in Hotan in 2050. In the context of China's strategic shift in population development, the study analyzed the demographic trends in ethnic minority areas and the impact of such trends on each region's social economy, resources, and environment. The purpose was to supply reference material for the formulation and implementation of population and economic policies in minority areas.

## 2. Adjustment of China's Population Policies and the Current Situation of Population Development in Hotan

Statistics released by the National Bureau of Statistics from the fifth population census in 2000, the sixth population census in 2010, and the sample survey data of the 1% population in 2015 indicated that the total fertility rate in China had fallen below 1.5 and kept declining. This suggested that China may have been caught in a "low-fertility trap" (a total fertility rate below 1.5) which would lead to labor shortages, fewer children, ageing, and other social problems that would be detrimental to China's long-term development. In 2016, the Chinese government released the "Two-Child" policy with the intention of encouraging childbearing. However, according to statistics released by the National Bureau of Statistics in early 2018, China had 630,000 fewer births in 2017 than in 2016 and the birth rate dropped by 0.52% to 12.43% (a figure even lower than Japan's birth rate). The natural population growth rate had fallen to 5.32%, an alarmingly low level. This indicated that the "Two-Child" policy failed to stem the downward trend in the fertility rate. In order to reach a total fertility rate of approximately 1.8 as planned, it will be necessary to abolish all birth restrictions.

Due to China's vast territory and numerous ethnic groups, attitudes towards fertility vary greatly among different regions. In terms of ethnic groups, the Uighurs accounted for 96.92% of the total population of Hotan in 2017. The family planning policy in ethnic minority areas is relatively loose in consideration of their traditions. Urban ethnic minorities are allowed to have two children, while rural ethnic minorities are allowed to have three, both much higher than the number allowed for the Han ethnicity. Influenced by traditional values and religions, however, the local people have a stronger desire to have children. Results of the field research show that 51% of the respondents want to have more than three children. This explains the frequent violation of family-planning directives and the consistently high fertility rate in this area. According to the yearbook of demography of the Xinjiang Uyghur Autonomous Region, from 2000 to 2017 the average birth rate in Hotan was 21.94%, far higher than the figure of 16.01% for the Autonomous Region and 12.4% for the entire country during the same period. Hotan's total population amounted to 2,522,800 in 2017. With a population density of 300.66 people per square kilometer, the population carrying capacity of the oasis stood at a very high level. However, due to the lagging economy, the secondary and tertiary industries were less capable of attracting labor—A large number stayed in rural areas. However, the oasis contained limited land resources. Since the 1990s, the growing rural labor force had no fields on which to work. In addition, it is difficult for the workforce to migrate across regions due to language barriers, differences in religion and life practices, and lack of professional skills. The joint effect of high fertility rates, low labor mobility, and lagging development has resulted in "Population barrier lake" in Hotan, which means the growth of the labor force is considerably faster than the growth of employment. The work report of the Hotan government showed that over 600,000 workers (which made up 40% of the local workforce) in Hotan had no jobs and no fields. Most of these people are young adults born in the 1980s and 1990s; they impose a great negative impact on the stability of society if they idle around. To provide arable land resources to the growing rural labor force, the area of arable land in Hotan increased from 2,574,000 mu in 2010 to 2,824,400 mu in 2017. The expansion of arable land put great pressure on regional resources. The Hotan region has exceeded the target maximum of total water consumption set by the central government. According to the 2010 Hotan census, the population aged under 45 accounted for 80.72% of the total population, with a median age of 25.04. Women of reproductive age (between 15 and 49) made up 29.24% of the total population, and women in their most fertile age (between 20 and 29) made up 11.05%. The region of Hotan has an expansive population pyramid which indicates that the growth rate will remain high for a long period. Excessive population growth and the resulting pressure on employment, resources, and the environment have become core constraints to the sustainable development of the Hotan region.

## 3. Materials and Methods

### 3.1. Data Sources

The data needed for the study are mainly from government data and field survey. Government data include Statistics released by the Xinjiang Bureau of Statistics from the sixth population census in 2010, Statistics from the 2010 Census of Hotan Prefecture, Statistical Yearbook of Hotan Prefecture from 2011 to 2018, and sample survey data of the 1% population in 2015. Survey data are the questionnaire survey conducted by the research team on the fertility intention and migration intention of the population in the Hotan region in 2016. The total number of questionnaires is 1036, mainly analyzing the local residents' fertility intention and migration intention.

### 3.2. Parameter Setting

PADIS-INT was applied to make projections on the population scale and age structure of the Hotan region in 2050. The software requires population projection parameters that include total fertility rate (TFR), life expectancy, gender ratio for new-borns, fertility patterns, and net inflow [28].

The total fertility rate is the key indicator for the projections; however, relevant statistics are available only from the decennial census. To discover the total fertility rate in Hotan from 2010 to 2017, different schemes of total fertility were set up with 2010 (data of 2010 The Sixth Population Census of Xinjiang Uygur Autonomous Region) as the starting year and 2017 (with the population data released in the 2017 Statistical bulletin on national economy and social development of the Hotan region) as the target year. The TFR was set to low and high based on fertility data from 2010 The Sixth Population Census of Xinjiang Uygur Autonomous Region and that of neighboring regions. Based on the calculation, the total fertility rate in Hotan during 2010–2017 was determined, based on which the demographic trend in Hotan in 2050 was projected.

The Low TFR: It was set at 1.8, the TFR from 2010 The Sixth Population Census of Xinjiang Uygur Autonomous Region.

The High TFR: Taking Pakistan (which was at a similar level of religious practice and economic development to the Hotan region) as a reference, the initial TFR in Hotan was set at 3.3, based on the 2011 total fertility rate data for Pakistan published by the United Nations Population Division.

Average life expectancy: Due to the lack of gender-specific life expectancy data for the Hotan region in 2010, the 2010 life expectancy in Xinjiang (70.30 for males and 74.86 for females) was used instead. The total fertility rate determined was used as a benchmark to calculate the annual gender-specific life expectancy for the Hotan region from 2010 to 2050 using the United Nations Life Expectancy calculation.

Gender Ratio of Newborns: According to The Hotan Region Statistical Yearbook, the mean gender ratio of the population of the region from 2001 to 2017 was 104.28 and the figure each year was relatively stable. Therefore, it was assumed that the gender ratio for newborns in the Hotan region was 104.28 during the period to be estimated.

Fertility patterns: There was a lack of data on age-specific fertility rates for women of reproductive age (15–49) in the Hotan region. The only available data were grouped by five years and were very close to those of the whole Xinjiang region (for rural areas). Therefore, the age-specific fertility rate in the Xinjiang region was taken to substitute the age-specific fertility rate in the Hotan region.

Migration: There was a lack of gender-specific demographic data on migration in the Hotan region. Since the changes in the total population of a region are caused by the combined effect of natural and mechanical changes, the number of mechanical changes can be calculated from the annual changes in the total population and the number of natural changes in the Hotan region and the number of mechanical changes was taken as the number of net migration. In order to improve the accuracy, the total population and the number of natural increases of the population from 2000 to 2017 were collected to calculate the number of migrations per year, which was positive at 13,307 as the average annual net migrant population in Hotan. Assuming a one-to-one gender ratio of the population

moving in each year and the number of people moving in during the period to be estimated remains unchanged, the number of migrant populations for each gender in the Hotan region during the period was 6653.

Migration mode: The ratio of the migrant population by age and by gender to the total migrant population was calculated according to the 2010 Hotan census.

*3.3. Total Fertility Rate and Setting Fertility Scenarios*

3.3.1. Determination of the Total Fertility Rate

The total fertility rate was set at 1.8 and 3.3, respectively with 2017 set as the target year. In the results given by PADIS-INT, the 2017 population estimated is 2,512,600 when the TFR is high. This is very close to the actual figure of 2,522,800 in 2017, with an error of only 0.4% (Table 1). The error between the low TFR and the real data is large, the error rate is as high as 8.82%, so it can be determined that 3.3 is the current total fertility rate in the Hotan area.

**Table 1.** Comparison of population estimated with the high and low total fertility rate (TFR) and the actual population in 2017.

|  | Low TFR | High TFR | Actual Population |
|---|---|---|---|
| Population in 2017 (ten thousand people) | 230.02 | 251.26 | 252.28 |

3.3.2. Setting Fertility Scenarios

Based on the data from the Sixth Population Census in Hotan, the total fertility rate was set at 3.3 as estimated in 2010–2017. Considering that the Chinese government may abolish the family planning policy in the near future, the total fertility rates in Hotan were set at three levels: Low, medium, and high. Other parameter settings remained unchanged, referring to the fertility rate changes in some Muslim countries released by the United Nations Population Division.

The Low TFR: As can be seen from the 2017 population, the total fertility rate in Hotan was comparable to that in Pakistan, where there is no family planning. It showed that the current family planning policy is of little constraint to local people, whose fertility intentions have been fully expressed. This indicated that even if the family planning policy was abolished, it would not bring about an obvious increase in local fertility intentions; this has already been proved by practices of China's hinterland provinces and those of Iran. Due to the socioeconomic development and the growing influence of modern values, the local fertility rate will continue to decline, assuming a gradual decline from the current 3.3 to 1.8 in 2050.

The Medium TFR: According to research in Hotan, 51% of the respondents think that the current three-child policy does meet their expectations; they want to have more children. Among those respondents, 66.9% wanted to have four children. In the meantime, 68.8% of the respondents expressed an intention to have fewer children due to the pressure of raising them and the harsh employment market. Thus, in the medium TFR scenario, it was assumed that the termination of family planning would not lead to a significant increase in fertility intentions: The total fertility rate in Hotan would rise slightly from the current 3.3 to 3.5 and then begin to decline until reaching 2.1 in 2050.

The High TFR: This assumes that the termination of family planning will greatly stimulate the local repressed intention to have more children. According to the survey, among the respondents who want to have more children, most of them choose to have four. Thus, this scenario assumes that the termination of family planning will lead to an increase in the local total fertility rate from 3.3 to 4.0 and thereafter gradually decrease to 2.1 in 2050 (Table 2).

**Table 2.** The low, medium, and high TFRs.

| Scenario | 2011–2019 | 2020–2029 | 2030–2039 | 2040–2050 |
|----------|-----------|-----------|-----------|-----------|
| Low | 3.3 | 3.0 | 2.5 | 1.8 |
| Middle | 3.3 | 3.5 | 2.8 | 2.1 |
| High | 3.3 | 4 | 3 | 2.1 |

## 4. Results and Analysis

### 4.1. Total Population

The PADIS-INT was applied to give projections with low, medium, and high TFR scenarios. The results of all scenarios showed that in the longer term, the population in Hotan would keep a steady growth. In the Low TFR Scenario, the total population of Hotan will grow from 2,010,000 in 2010 to 4,430,700 in 2050, with an average annual growth rate of 1.99%. In the Medium TFR Scenario, the total population of Hotan will reach 4,760,900, with an average growth rate of 2.17%. In the High TFR Scenario, the total population of Hotan will reach 4,958,600 in 2050 (Table 3), with an average annual growth rate of 2.28%.

**Table 3.** The total population projection in the Hotan Region (10,000 people).

| Year | Low TFR Scenario | Medium TFR Scenario | High TFR Scenario |
|------|------------------|---------------------|-------------------|
| 2020 | 272.17 | 273.22 | 274.27 |
| 2030 | 333.36 | 344.24 | 354.91 |
| 2040 | 392.11 | 411.02 | 426.58 |
| 2050 | 443.07 | 476.09 | 495.86 |

### 4.2. The Population Structure

#### 4.2.1. The Age Structure of Population and the Trend

The conclusions can be drawn from the population age pyramid in 2050 that regardless of scenario, the population age structure of Hotan will appear pyramidal (Figure 1). Moreover, the population under age 40 takes up a large proportion: 59.95%, 62.73%, and 64.21% of the total population in terms of the low, medium, and high TFR scenarios, respectively—with the median age being 33.18, 31.05, and 29.81. Compared to the median age of 37 among Chinese people in 2015 and 49.6 in 2050, the age structure of the population in the Hotan region appears extremely young (Figure 2).

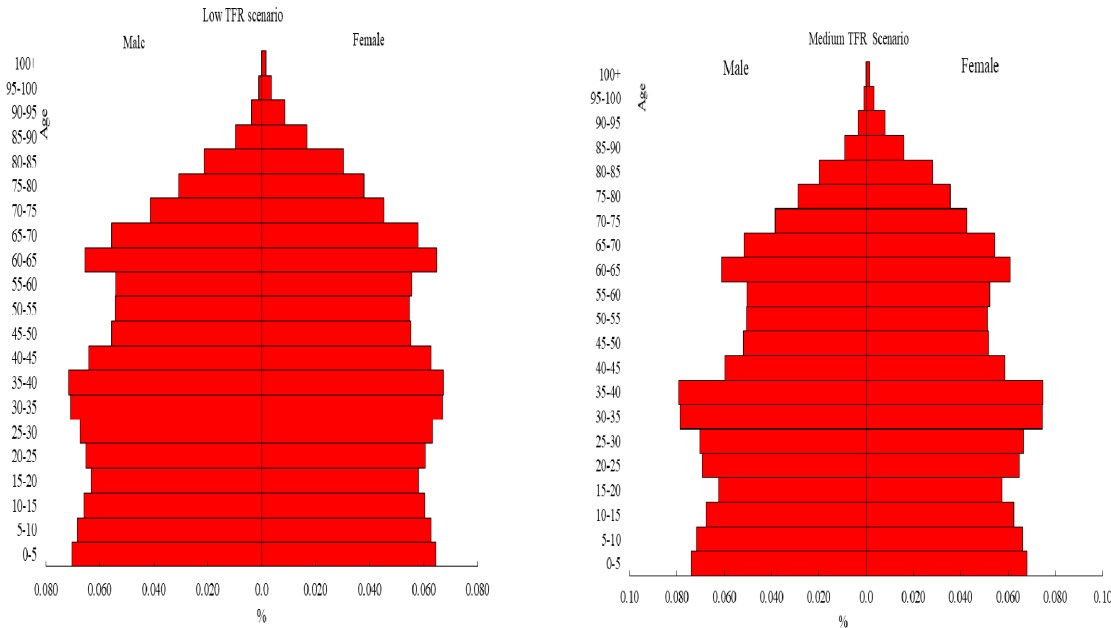

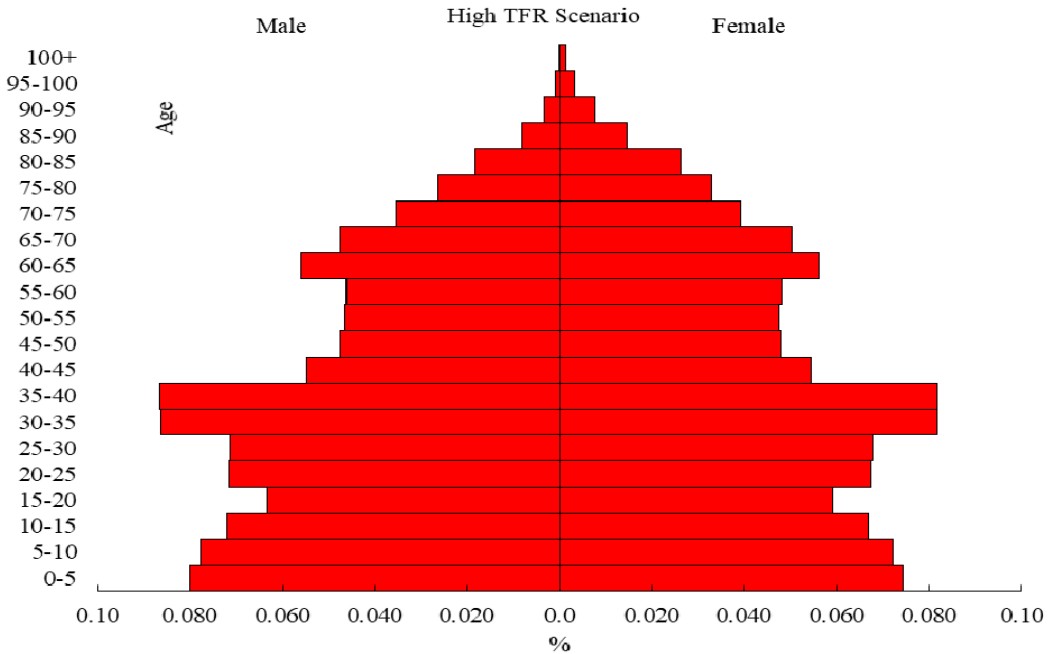

**Figure 1.** The population age pyramids of the Hotan region in 2050.

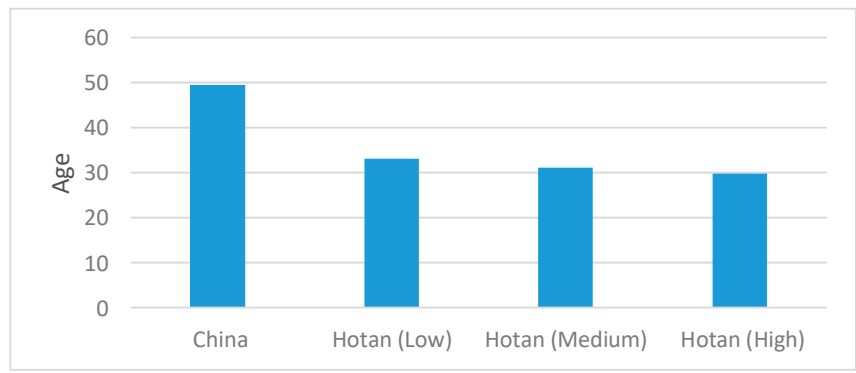

**Figure 2.** Comparison of the medium ages in 2050.

### 4.2.2. Changes in the Labor Force

The increasingly youthful age structure of the population produces a large workforce. Internationally, the working age is generally defined as those aged 15–64. According to the projections, the labor force in the Hotan region for the low, medium, and high TFR scenarios will reach 2,874,600, 3,017,700, and 3,147,000, respectively by 2050. Compared to the figure of 1,398,300 in 2010, the labor force in either scenario will more than double by 2050 (Figure 3), taking up between 63% and 64% of the total population. More than 600,000 of the labor force are now unemployed with no arable land, as local natural resources are limited. If secondary and tertiary industries are not maintained at a high rate of growth or the interregional flow of the workforce is not promoted, the number of unemployed people will double, posing enormous pressure to social stability.

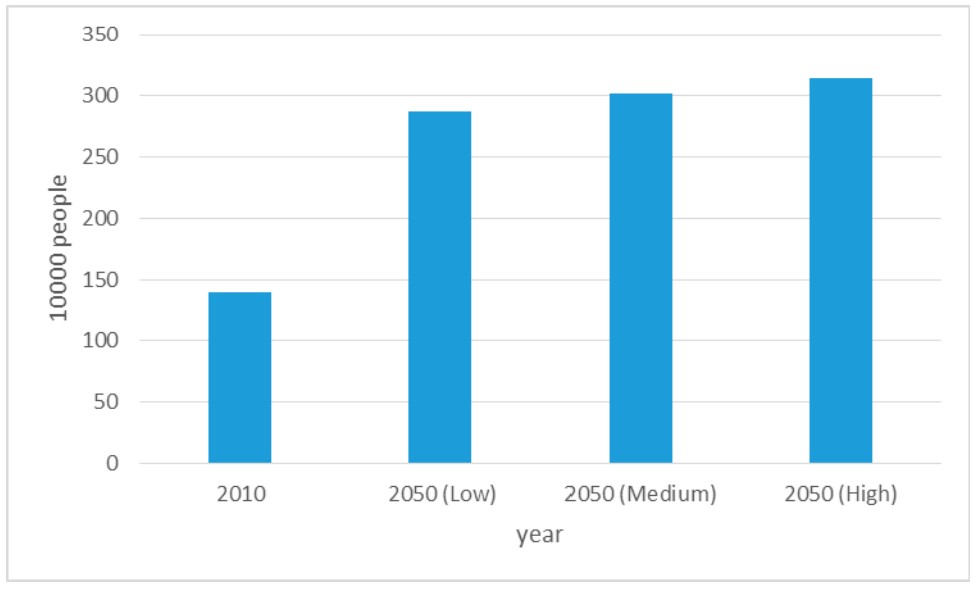

**Figure 3.** Comparison of the number of labor force in Hotan.

### 4.2.3. The Number of Women of Reproductive Age and Trends

The number of women of reproductive age affects the future demographic trend. The number of women of reproductive age in Hotan will increase from 603,000 in 2010 to between 1,084,900 and 1,218,800 in 2050. Among them, the number of women in the peak reproductive age (age 20–29) will increase from 222,300 to between 329,800 and 425,100 (Figure 4). The women of childbearing age in all three scenarios accounted for around 24% of the total population, with a decrease of 5% (from 29.94%) in 2010. Meanwhile, the proportion of women of childbearing age ranged between 7.44% and 8.57%,

with a decrease of 3–4% compared to 2010. Despite this decline, the absolute number of women of childbearing age will nearly double in the next three decades.

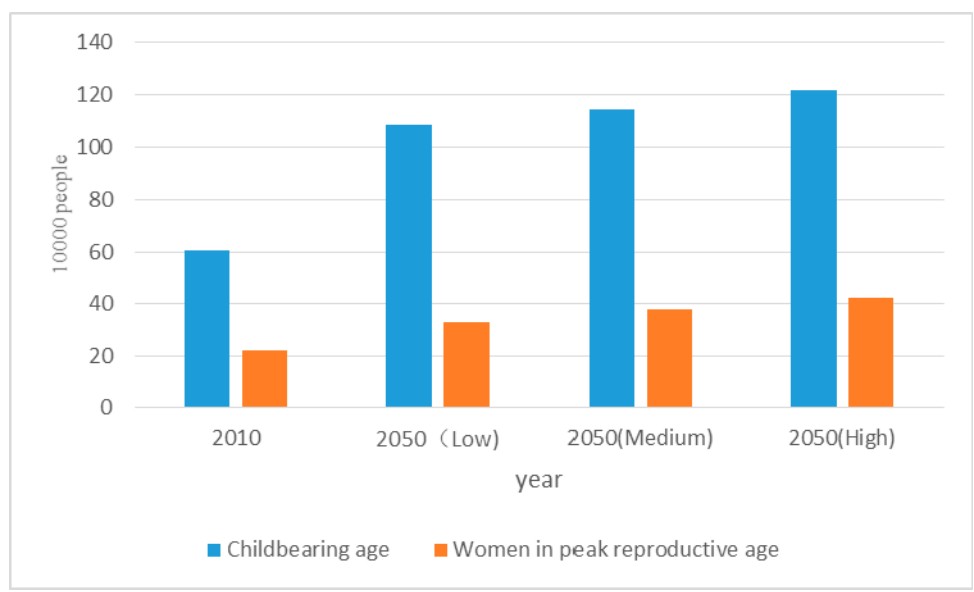

**Figure 4.** Change in the number of women of reproductive age by 2050.

### 4.2.4. Social Dependency Ratio and Trends

Following international standards, the child dependency ratio is defined as the number of the population aged 0–14 to the population aged 15–64. The old-age dependency ratio is defined as the number of the population aged 65 and above to the population aged 15–64 years. The total dependency ratio is the sum of the child dependency ratio and the old-age dependency ratio. According to the projection results, the population aged 0–14 in Hotan will increase from 528,300 in 2010 to between 930,500 and 1,185,900 in 2050 and the child dependency ratio will increase from 37.78% in 2010 to between 54.13% and 57.77% in 2050. The number of people aged over 65 is influenced by high fertility rates in the early period and will rise sharply from 87,700 in 2010 to 625,600 in 2050, an increase of 7.13. The old-age dependency ratio will rise from 6.27% to 19.88–21.76%. The elderly-youth ratio, influenced by the increase in the elderly population, will increase from 16.6% in 2010 to between 52.75% and 67.23% in 2050. The social dependency ratio increased from 44.05% in 2010 to 74.01–79.53% in 2050, promoted by the rise of the old-age dependency ratio.

In all scenarios, comparing the population structure of Hotan in 2050 based on the United Nations classification criteria of population structure types (Table 4), the Hotan region has already entered into an ageing society in 2050. However, the median age of the Hotan region in the three scenarios is 33.18, 31.05, and 29.81, respectively. Comparing those results with the United Nations Population Division's estimates on the median ages of various countries in 2050, the median age of the Hotan region in 2050 is not only much lower than that of China (49.6) but also much lower than that of Muslim countries such as Iran and Saudi Arabia—and on par with that of Afghanistan [29] (Table 5). Therefore, due to the influence of rapid population growth, the Hotan region will face pressure from both a tough employment situation and the heavy burden of social support in the near future. However, the proportion of the workforce in the total population will stay between 63% and 64%, with a median age of around 30. Although the Hotan region will have already entered into an ageing society by the current criteria, its population structure is still relatively youthful compared to the population structures of other countries in 2050. Although the Hotan burden of social support will rise in the near future, the principal contradiction in society will still be focused around the employment problem of a large number of workers brought about by the population growth.

**Table 4.** Categorization criteria of the population structure.

|  | Old-Age Ratio | Child Ratio | Elderly-Youth Ratio | MEDIAN AGE |
|---|---|---|---|---|
| Expansive | Below 4% | Below 40% | Below 15% | Below 20 |
| Constructive | 4–7% | 30–40% | 15–30% | 20–30 |
| Ageing | Over 7% | Below 30% | over 30% | over 30 |

**Table 5.** Median age in 2050.

| China | China (Hotan) | | | Saudi | Indonesia | Pakistan | Afghanistan | Iran |
|---|---|---|---|---|---|---|---|---|
|  | Low Scenario | Medium Scenario | High Scenario |  |  |  |  |  |
| 49.6 | 29.81 | 31.05 | 33.18 | 38.2 | 36.5 | 30.9 | 29.8 | 44.7 |

## 5. Discussion

The complete lifting of birth restrictions is to take place in the near future. Due to the expansive population structure of Hotan, its population growth area will see a rapid increase after the abolition of the family planning policy. Located on the southern edge of the Taklamakan Desert, the Hotan region is an arid and ecologically fragile area. In order to protect the environment, the Chinese government has set a limit on the total amount of water used in each region, known as "the red line for water resources". However, the local government has continued to increase the development of land and water resources to meet the needs of the growing workforce. The area of arable land increased from 2,574,000 mu in 2010 to 2,824,400 mu in 2017. However, due to the rapid population growth, the arable land per capita was only 1.12 mu. Moreover, the expansion of arable land has already put tremendous pressure on regional resources and the environment. The exploitation rate of water resources in Hotan was as high as 79.86%, 98% of which was taken up by agricultural use. Internationally, it is generally believed that the development and utilization of water resources should not exceed 40% of its water resources. The threshold of water resources development and utilization of China's seven major rivers is 31% to 45%, and for relatively fragile ecological environment in the northwest arid region, the proportion of resource development utilization rate should not exceed 50% [30], and the utilization rate of Hotan water resources development is much higher than the reasonable level. The total water consumption in Hotan in 2015 had already exceeded the national water consumption control target by 2030 by nearly 700 million cubic meters, which indicated that the expansion of arable land could no longer continue.

Since there is no room for expanding arable land, the main solution to the employment problem is to transfer the workforce from the primary industry to secondary and tertiary industries. In 2017, the ratio of the primary, secondary, and tertiary industries in the Hotan region was 22.28:17.9:59.8. In terms of the output value, the secondary industry's output was lower than that of the primary industry. The urbanization rate in 2017 was merely 19.77% and the GDP per capita was RMB 10,400. Agriculture was the pillar of industrial development, absorbing up to 68.2% of the labor force. The Hotan region was still in the stage of transformation from an agricultural society to early industrialization. The Hotan region has been developing textiles and other labor-intensive industries in recent years to boost employment capacity. There was an increase in the employment elasticity coefficient, which reached 1.08 in 2017. Assuming that the employment elasticity can be maintained over the long term by excluding the declining effects brought about by technology and capital substitution, if the Hotan region seeks full employment its economic growth rate should be at least 9.55% for the next three decades (to meet the amount of newly increased workforce estimated in whichever scenario of population projection). Compared to China's development since the reform and opening up, such a growth rate may not seem too difficult in the early stages; however, as the total volume of the economy increases, the growth rate will inevitably be slow. Considering that the industrial base of the Hotan region and its geographical location are far from China's major consumer markets, it is basically impossible to maintain such high growth for the next three decades. If economic

growth does not achieve the expected speed, the number of unemployed and landless workforce could surpass one million.

It is difficult to rely entirely on the Hotan region to solve the employment problem for the newly added workforce. The ultimate way to solve the problem is by cross-regional employment, which is to encourage the outflow of the local labor force to well-developed areas. However, Hotan is an ethnic minority region. Due to many restrictions including the language barrier, great differences in living habits and religious customs, strong emotional attachment to the hometown, and lack of professional skills, Hotan does not see a great deal of mobility among its population. Local residents are generally unwilling to migrate. Even for those who choose to work outside their hometown, the Hotan region continues to be their home. Among the more than 1000 residents surveyed, 91.32% were unwilling to move outside Hotan for employment. The relatively closed geographic and social environment in Hotan makes it difficult to achieve a large-scale population mobility. In order to promote the population mobility while respecting unique cultural customs, it would be appropriate to establish Uyghur gathering areas in other cities (similar to other countries' designations of "Chinatowns"). By creating a homelike living environment, the Uyghur people's worries can be alleviated and they may be more comfortable seeking employment in other provinces.

The development of education is the fundamental strategy to solve the contradiction between population and resource environment in Hotan. According to statistics from the Sixth Population Census in Hotan, 89.06% of the workforce has only a junior high school education or less. The National Family Planning Commission's survey on China's fertility status shows that when women have been educated for more than 7 years, the average number of children born per person is 2.2 fewer than those without education. Low academic qualifications not only increase the fertility rate, but also make it difficult for the huge population to turn into a "demographic dividend", but it has become a heavy social burden. Raising educational standards to prepare a more qualified workforce is a premise for changing the region's lagging development. The Hotan region should further invest in education to build a system that comprises compulsory, vocational, adult, and higher education. By improving the professional skills of the workforce, it will lay a foundation for the outward mobility of the workforce while training talents for local socioeconomic development.

## 6. Conclusions

Against the background of China's relaxation of family planning standards, this thesis analyzed the demographic trends in ethnic minority areas and their impacts on regional development under China's adjustment of its population strategy. The results showed that the Hotan region would maintain a relatively fast population growth rate for a long period; by 2050, its population would skew younger when compared to China's general statistics. However, due to the lagging economic development and the constraints to resources and the environment, the "Population barrier lake" in Hotan may be more severe [4]. Due to the lack of data, this paper is not in-depth enough to analyze the impact of population flow and population internal structure changes on the regional resources and the environment. In the future, it is necessary to strengthen data collection, establish long-term population structure data, and systematically analyze the relationship between population and resources and environment.

**Author Contributions:** J.H. conceived and designed this study; X.Z. retrieved and analyzed the data; Z.Z. contributed analysis software; J.H. wrote the paper; Y.C. revised the paper. All authors have read and agreed to the published version of the manuscript.

**Funding:** This work was supported by the Light project of the West (2019-XBQNXZ-A-005).

**Conflicts of Interest:** The authors declare no conflict of interest.

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
