# Peer review of "Research on Population Development in Ethnic Minority Areas in the Context of China’s Population Strategy Adjustment"

_sustainability, doi:10.3390/su12198021_

Round 1

Reviewer 1 Report

  • In the discussion of the paper the authors state: “Improve the education level in ethnic minority áreas”. It would be interesting to cross these data with those of other areas where high educational level may be related to low natality rate, especially if it is a question of women reaching educational levels that allow them to access the labor market with a more complete education.
  • Also the relationship between the mobility of the population to other countries and the relationship with the lower natality rates among this population.

Author Response

1、In the discussion of the paper the authors state: “Improve the education level in ethnic minority áreas”. It would be interesting to cross these data with those of other areas where high educational level may be related to low natality rate, especially if it is a question of women reaching educational levels that allow them to access the labor market with a more complete education.

The Chinese government didn't release related data to conduct this study, but we cite a government report that shows a strong correlation between education and fertility.(11 of 12, 5. Discussion The National Family Planning Commission’s survey on China’s fertility status shows that when women have been educated for more than 7 years, the average number of children born per person is 2.2 fewer than those without education.)

2、Also the relationship between the mobility of the population to other countries and the relationship with the lower natality rates among this population.

    There are very few floating populations in Hotan, and the government has not released relevant data.

Reviewer 2 Report

The article is interesting and up to date, but I am not sure if it is touching the problem within the topic of the Journal.. There is no theoretical background shown at all.
There is no objective mentioned in the abstract.
The contribution of the paper is rather poorly explained. The authors do not concentrate to show the gap in literature they want to fulfill. The literature review is also ather poor and should be improved especially when it comes to discussion of the results which is in fact missing.
The language should be improved.
The references are edited not in line with the Journal guidelines
All the abbrevations should be explained in the text and in the figures descriptions because the structure of the article is not clear enough.

Author Response

1、There is no objective mentioned in the abstract.

Has increased(The fifth line of Abstract :This thesis analyzed the impacts of population growth on regional sustainable development from the perspectives of employment, economic development and resource carrying capacity to provide references for the formulation and implementation of population and economic development policies in minority areas, to alleviate the contradiction between the human and environment.)

2、The contribution of the paper is rather poorly explained. The authors do not concentrate to show the gap in literature they want to fulfill. The literature review is also ather poor and should be improved especially when it comes to discussion of the results which is in fact missing.。

We have revised the introduction based on your comments.(1. Introduction In the context of global change, with the increase in the size of the population and its material and cultural needs, the intensity of human activities and their impact on the society, economy and ecosystems have increased, and the internal conflicts in the human-environment relationship have become increasingly prominent [1]. The correlation between population and resources and environment is a central issue for regional sustainable development [2-4]. Humankind has long recognized that the Earth has limited space and resources. Humans have also realized the importance of living in harmony with nature, making great efforts to learn about their surrounding resources and environment [5]. The survival and development of human beings need to obtain sufficient means of production from the resource and environment system. However, under certain technological conditions, the demand of human beings exceeds the supply of the resource and environment system, which will lead to the deterioration of the environmental system and endanger the survival and development of human beings. Maintaining the population within the carrying capacity becomes the scale to measure the state of regional sustainable development. The Malthusian theory of population [6], Schultz’s human capital theory [7] and the demographic dividend theory of Bloom et al are all concerned with the effects of population growth externally [8]. The Limits to Growth, published in 1972 by the Club of Rome, first defined resource and environmental carrying capacity: rapid industrialization, soaring population growth, food privatization, depletion of non-renewable resources and degradation of the ecosystem. At some point, global growth will hit its limit due to food shortages and ecological devastation. That limit is the carrying capacity [9]. As its population continues to grow and socioeconomic development accelerates, problems such as resource shortages and environmental pollution have become increasingly important constraints to sustainable development in some parts of China. China has launched a series of scientific and technological projects targeting the north-west and the north-east regions to study supporting capability and carrying capacity of resources and environment for each region’s society and economy [10-16]. Studies have also been conducted on the supporting capabilities and protection levels of the resources and environment for each region’s population and economy, point out that the increased vulnerability of the environmental system caused by the increase in human activities has become the main obstacle to the sustainable development of the regional ecological environment and social economy, and propose measures to implement ecological migration, develop carbon sink industry, and protect ecological water, The goal is to provide a scientific basis for the realization of regional resources and environmental security as well as the sustainable development of the population’s socioeconomic conditions[17-21].

To maintain the balance between population and resource environment, China has a long history of prioritizing population control. Since the implementation of family planning policies, 400 million fewer people have been born in China. This has effectively alleviated the pressure population imposes on the environment and resources; however, the long-term low fertility rate has led to an ageing problem. Even after the Chinese government relaxed birth restrictions, the total fertility rate (TFR)continued to show a downward trend; An excessively low fertility rate can have a negative impact on socioeconomic development. As China’s socioeconomic development enters a new era, China has shifted its population strategy from merely controlling the number to addressing specific issues (including population scale, structure and quality) in a coordinated manner. The population strategy has also been adjusted from addressing internal problems of the population system to rational interactions with the external environment [22]. However, situations in ethnic minority areas are different from those in China as a whole. Due to preferential policies for ethnic minorities, family planning is already relatively loose in ethnic areas with relatively rapid population growth. Moreover, since most of China’s ethnic minority regions are located in impoverished places (such as the frontier and drought-stricken areas) where economic development lags and productivity levels are low, increasing the labour force has been the main way out of poverty. Inhabitants in ethnic minority areas generally share a strong will to have children which has resulted in a population growth rate much higher than the national average. However, when the total population exceeds the carrying capacity of the region’s resources and environment, it starts to cause unemployment and environmental crises. The population problem has become a critical constraint for the sustainable development of China’s ethnic areas and the realization of poverty alleviation. However, there are few studies on the impact of the adjustment of family planning on population development in ethnic minority areas. On the southern edge of the Taklamakan Desert, Hotan in Xinjiang features a fragile ecological environment; it is a minority area located on the border and a contiguous area of extreme poverty. It is a typical ‘frontier, minority and impoverished’ region in China. This case study used the example of the ethnic minority area of Hotan, The hypothesis was the abolishment of the family planning policy. By setting up different fertility scenarios, the Population forecasting software (PADIS-INT) was applied to forecast the population scale and structure in Hotan in 2050. In the context of China’s strategic shift in population development, the study analyzed the demographic trends in ethnic minority areas and the impact of such trends on each region’s social economy, resources and environment. The purpose was to supply reference material for the formulation and implementation of population and economic policies in minority areas.)

3、The language should be improved.

The article was translated by Elsevier

4、The references are edited not in line with the Journal guidelines

The modified

5、All the abbrevations should be explained in the text and in the figures descriptions because the structure of the article is not clear enough.

The structure of the article has been modified as required, and the abbreviation has been marked on the first occurrence(The Population forecasting software (PADIS-INT),Total fertility rate (TFR))

Reviewer 3 Report

Dear Authors,

The submitted manuscript titled „Research on Population Development in Ethnic Minority Areas in the Context of China’s Population Strategy Adjustment - Exemplified by the Hotan Region in Xinjiang” contains interesting results. However, I found some flaws, which must be corrected before eventual publication.

  1. Title reflects the content of manuscript but in my opinion it is too long and too descriptive.
  2. Chapter “Introduction” presents the background of the studies but in my opinion there is lack of information if similar investigations were carried out previously in ethnic minority living in China. Moreover, You should explain why investigations were carried out in Hotan. Furthermore, there is lack of specific goals of investigations, which should be listed “one by one” at the end of chapter.
  3. Chapters “Material and Methods” and “Results” should be established. The chapter “Material and methods” should contain information about way of data collection, period of investigations, parameter characteristic. I suggest to resign from short subchapters devoted to particular parameters. Chapter “Results” should contain outcomes and must be more concise. Please, avoid repetitions of data from Tables in the text.  

In my opinion subchapter “Impact of Population Growth on Regional Development” should be moved into “Discussion” section.

  1. In “Discussion” section there is lack of comparisons of obtained results with literature of subject.
  2. I suggest to look into below listed publications, which perhaps might be helpful in improvement of manuscript:

Luo et al. 2020. Chinese trends in adolescent marriage and fertility between 1990 and 2015: a systematic synthesis of national and subnational population data. LANCET GLOBAL HEALTH  8(7): E954-E964.

Han et al. 2020. Evaluation of human-environment system vulnerability for sustainable development in the Liupan mountainous region of Ningxia, China. ENVIRONMENTAL DEVELOPMENT 34: Article Number: UNSP 100525. DOI: 10.1016/j.envdev.2020.100525

Belando-Montoro, MR, Zhou, ZQ 2020. A research on education of minority female in China's Xinjiang region. ATHENEA DIGITAL 20(1): Article Number: e2124. DOI: 10.5565/rev/athenea.2124

Soyoung 2019. The Reform and Open Policy of China and the Development of Xinjiang Uyghur Autonomous Region. DAEGUSAHAK 135: 227-263. DOI: 10.17751/DHR.135.227 

Author Response

  1. Title reflects the content of manuscript but in my opinion it is too long and too descriptive.

The title has been modified as “Research on Population Development in Ethnic Minority Areas in the Context of China’s Population Strategy Adjustment “

  1. Chapter “Introduction” presents the background of the studies but in my opinion there is lack of information if similar investigations were carried out previously in ethnic minority living in China. Moreover, You should explain why investigations were carried out in Hotan. Furthermore, there is lack of specific goals of investigations, which should be listed “one by one” at the end of chapter.

According to your opinion, we revise as follows:(1. Introduction  On the southern edge of the Taklamakan Desert, Hotan in Xinjiang features a fragile ecological environment; it is a minority area located on the border and a contiguous area of extreme poverty. It is a typical ‘frontier, minority and impoverished’ region in China. This case study used the example of the ethnic minority area of Hotan, The hypothesis was the abolishment of the family planning policy. By setting up different fertility scenarios, the Population forecasting software (PADIS-INT) was applied to forecast the population scale and structure in Hotan in 2050. In the context of China’s strategic shift in population development, the study analyzed the demographic trends in ethnic minority areas and the impact of such trends on each region’s social economy, resources and environment. The purpose was to supply reference material for the formulation and implementation of population and economic policies in minority areas.)

3 of 12, 3.1. Data Sources Survey data is the questionnaire survey conducted by the research team on the fertility intention and migration intention of the population in Hotan region in 2016. The total number of questionnaires is 1036, mainly analyzing the local residents' fertility intention and migration intention.)

  1. Chapters “Material and Methods” and “Results” should be established. The chapter “Material and methods” should contain information about way of data collection, period of investigations, parameter characteristic. I suggest to resign from short subchapters devoted to particular parameters.

Chapters “Material and Methods” and “Results” have been established ( 3. Materials and Methods. 4. Results and Analysis) .

The chapter “Material and Methods” has been modified as suggested “Material and methods”(3 of 12, 3.1. Data Sources  The data needed for the study are mainly from government data and field survey. Government data include Statistics released by the Xinjiang Bureau of Statistics from the sixth population census in 2010, Statistics from the 2010 Census of Hotan Prefecture, Statistical Yearbook of Hotan Prefecture from 2011 to 2018 and sample survey data of the 1% population in 2015. Survey data is the questionnaire survey conducted by the research team on the fertility intention and migration intention of the population in Hotan region in 2016. The total number of questionnaires is 1036, mainly analyzing the local residents' fertility intention and migration intention.).

The short sub-chapters have been merged (4 of 12, 3.2. Parameter Setting).

  1. Chapter “Results” should contain outcomes and must be more concise. Please, avoid repetitions of data from Tables in the text.  

We have deleted the repetitions.(6 of 12 , 4.1. Total population The PADIS-INT was applied to give projections with low, medium and high TFR scenarios. The results of all scenarios showed that in the longer term, the population in Hotan would keep a steady growth. The Low TFR Scenario, the total population of Hotan will grow from 2,010,000 in 2010 to 4,430,700 in 2050, with an average annual growth rate of 1.99%. The Medium TFR Scenario, the total population of Hotan will reach 4,760,900, with an average growth rate of 2.17%. The High TFR Scenario, the total population of Hotan will reach 4,958,600 in 2050 (Table 3), with an average annual growth rate of 2.28%.)

  1. In my opinion subchapter “Impact of Population Growth on Regional Development” should be moved into “Discussion” section.

The modified (5. Discussion)

  1. In “Discussion” section there is lack of comparisons of obtained results with literature of subject.

The modified(10 of 12,Line 12 of the first paragraph of discussion The exploitation rate of water resources in Hotan was as high as 79.86%, 98% of which was taken up by agricultural use. Internationally, it is generally believed that the development and utilization of water resources should not exceed 40% of its water resources. The threshold of water resources development and utilization of China’s seven major rivers is 31% to 45%,and for relatively fragile ecological environment in the northwest arid region, the proportion of resource development utilization rate should not exceed 50%)

  1. I suggest to look into below listed publications, which perhaps might be helpful in improvement of manuscript:

Thank you for your suggestions. We have read these publications and listed some of them for reference(Han, X.J.; Wang, P.; Wang, J.J.; Qiao, M.; Zhao, X.C. Evaluation of human-environment system vulnerability for sustainable development in the Liupan mountainous region of Ningxia, China. Environmental Development 2020, 34.

Round 2

Reviewer 2 Report

The authors corrected the article in line with the previous comments. I suggest to publish the article in the present version.

Author Response

Thank you reviewers

Reviewer 3 Report

Dear Authors,

The submitted manuscript has been improved, therefore I do not have any further suggestions.

Author Response

Thank you reviewers